# Improved performance and consistency of deep learning 3D liver segmentation with heterogeneous cancer stages in magnetic resonance imaging

Moritz Gross[1,2], Michael Spektor[1], Ariel Jaffe[3], Ahmet S. Kucukkaya[1,2], Simon Iseke[1,4], Stefan P. Haider[1,5], Mario Strazzabosco[3], Julius Chapiro[1], John A. Onofrey[1,6,7] *

1 Department of Radiology and Biomedical Imaging, Yale University School of Medicine, New Haven, Connecticut, United States of America, 2 Charité Center for Diagnostic and Interventional Radiology, Charité —Universitätsmedizin Berlin, Berlin, Germany, 3 Department of Internal Medicine, Yale University School of Medicine, New Haven, Connecticut, United States of America, 4 Department of Diagnostic and Interventional Radiology, Pediatric Radiology and Neuroradiology, Rostock University Medical Center, Rostock, Germany, 5 Department of Otorhinolaryngology, University Hospital of Ludwig Maximilians Universität München, Munich, Germany, 6 Department of Urology, Yale University School of Medicine, New Haven, Connecticut, United States of America, 7 Department of Biomedical Engineering, Yale University, New Haven, Connecticut, United States of America

* john.onofrey@yale.edu

## Abstract

### Purpose

Accurate liver segmentation is key for volumetry assessment to guide treatment decisions. Moreover, it is an important pre-processing step for cancer detection algorithms. Liver segmentation can be especially challenging in patients with cancer-related tissue changes and shape deformation. The aim of this study was to assess the ability of state-of-the-art deep learning 3D liver segmentation algorithms to generalize across all different Barcelona Clinic Liver Cancer (BCLC) liver cancer stages.

### Methods

This retrospective study, included patients from an institutional database that had arterial-phase T1-weighted magnetic resonance images with corresponding manual liver segmentations. The data was split into 70/15/15% for training/validation/testing each proportionally equal across BCLC stages. Two 3D convolutional neural networks were trained using identical U-net-derived architectures with equal sized training datasets: one spanning all BCLC stages ("All-Stage-Net": AS-Net), and one limited to early and intermediate BCLC stages ("Early-Intermediate-Stage-Net": EIS-Net). Segmentation accuracy was evaluated by the Dice Similarity Coefficient (DSC) on a dataset spanning all BCLC stages and a Wilcoxon signed-rank test was used for pairwise comparisons.

### Results

219 subjects met the inclusion criteria (170 males, 49 females, 62.8±9.1 years) from all BCLC stages. Both networks were trained using 129 subjects: AS-Net training comprised

**Data Availability Statement:** Trained models and evaluation results can be found on GitHub at the following link: https://github.com/OnofreyLab/liver-

segm/. Image data used in this paper cannot be shared publicly due to legal reasons (it would compromise patient confidentiality). Data are available from the Yale Joint Data Analytics Team (JDAT) (contact researchdatarequest@yale.edu) for researchers who meet the criteria for access to confidential data.

**Funding:** J.A.O. was supported by the National Institute of Diabetes and Digestive and Kidney Diseases of the National Institutes of Health under Award Number P30 KD034989 and M.S. the National Institutes of Health Grant Award Number DDRCC DK034989-36 for the Clinical Translational Core of the Yale Liver Center. The content is solely the responsibility of the authors and does not necessarily represent the official views of the Nation Institutes of Health. M.G. was supported by a travel grant by the Rolf W. Günther Foundation for Radiological Sciences for travel to Yale University. The funders had no role in study design, data collection and analysis, decision to publish, or preparation of the manuscript.

**Competing interests:** The authors have declared that no competing interests exist.

19, 74, 18, 8, and 10 BCLC 0, A, B, C, and D patients, respectively; EIS-Net training comprised 21, 86, and 22 BCLC 0, A, and B patients, respectively. DSCs (mean±SD) were 0.954±0.018 and 0.946±0.032 for AS-Net and EIS-Net (p<0.001), respectively. The AS-Net 0.956±0.014 significantly outperformed the EIS-Net 0.941±0.038 on advanced BCLC stages (p<0.001) and yielded similarly good segmentation performance on early and intermediate stages (AS-Net: 0.952±0.021; EIS-Net: 0.949±0.027; p = 0.107).

## Conclusion

To ensure robust segmentation performance across cancer stages that is independent of liver shape deformation and tumor burden, it is critical to train deep learning models on heterogeneous imaging data spanning all BCLC stages.

## Introduction

Liver cancer is the third most common cause of cancer-related death worldwide [1] and both incidence rates and mortality are rising [2, 3]. Hepatocellular carcinoma (HCC) is the most prevalent form of primary liver cancer, accounting for 70–85% of liver cancers globally [4]. Magnetic resonance (MR) imaging offers high tissue contrast and with the use of contrast agents and multiphasic imaging, HCC can be detected and diagnosed reliably without the need for an invasive biopsy in a majority of cases [5]. Multiple staging systems have been developed to assess the stage of HCC and to provide guidance regarding optimal therapeutic management [6–10]. In particular, the Barcelona Clinic Liver Cancer (BCLC) staging classification [6] is widely accepted and the most commonly used in Western cohorts. The BCLC classification utilizes three clinical elements: tumor burden, functional status as measured by the Eastern Cooperative Oncology Group (ECOG) Performance Status [11], and underlying liver function measured by the Child-Pugh class [12] to stratify patients into five staging categories: very early stage (BCLC-0), early stage (BCLC-A), intermediate stage (BCLC-B), advanced stage (BCLC-C), and terminal stage (BCLC-D).

Accurate organ segmentation plays an important role in medical image analysis tasks. Liver segmentation is key for volumetry prior to therapeutic interventions [13–17] and as a pre-processing step for subsequent cancer detection algorithms [18, 19]. Accurate volumetry assessment is imperative to understanding the risk of hepatic decompensation associated with various treatment approaches and plays a critical role in management decisions. It has been shown that the critical residual liver volume necessary to prevent post-hepatectomy liver failure in non-cirrhotic patients is 20–30%, compared to at least 40% residual volume in cirrhotic patients. Thus, possible curative therapies again rely heavily on accurate volume assessment in patients with liver cancer [20]. However, manual segmentation is time-consuming and dependent on the rater's level of experience, which leads to a lack of reproducibility and inter-observer variability [21]. Heterogeneity in terms of disease stage and imaging appearance further complicates segmentation. Liver segmentation can be especially challenging in patients with abnormal liver function and significant disease complexity. Various morphologic changes occur in cirrhotic patients including left lobe hypertrophy, increased nodularity of the liver surface, portal hypertension often manifesting with significant ascites and changes in vasculature in addition to cancer-related tissue changes that alter the liver contour all contribute to substantial variations in the imaging morphology [22, 23]. In this paper, we use the BCLC

classification as a marker for liver function, severity of HCC, disease complexity, and overall imaging heterogeneity.

To improve liver segmentation reproducibility, automated methods based on image analysis methods and machine learning have been developed and shown promising results [24–27]. Current state-of-the-art methods utilize deep learning based on convolutional neural networks (CNNs) [28]. Such CNNs have demonstrated superior segmentation results across a wide variety of medical image segmentation applications [29] and also have the advantage of processing times in the order of seconds. In particular, these algorithms have been applied to segment the liver on computed tomography (CT) and MRI data [30–42]. However, machine learning algorithms, and in particular high-dimensional and non-linear deep learning algorithms, are prone to over-fitting, which results in models that are not robust to data that varies substantially from their training data [43]. This is a problem of distributional shift, or dataset shift, where a mismatch between distributions of training data and testing data exists [44]. Software development specifications aimed at ensuring quality in the development and the use of AI modules identify distributional shift as one of the major risks to robust application of AI [45]. To avoid distributional shifts caused by sample selection bias, it is critical that algorithms be trained on data representative of the test set.

Therefore, deep learning liver segmentation algorithms trained only on early and intermediate HCC stages will result in algorithms tuned to this specific patient population and thus fail to generalize to more advanced stages due to their heterogeneous imaging morphology. The aim of this study was to assess the ability of state-of-the-art deep learning 3D liver segmentation algorithms to generalize across the clinical distribution all different BCLC liver cancer stages.

## Materials and methods

### Inclusion of patients

This HIPAA-compliant, retrospective, single-institution study was IRB-approved with full waiver of consent and included all patients from an institutional database with T1-weighted arterial-phase MR images and a corresponding manual liver segmentation available for processing. All patients were >18 years old and had treatment-naïve HCC that was either imaging- or histopathologically-proven. Patient data was collected from the hospital's electronic health record and all patients were retrospectively staged according to the BCLC staging system.

### Magnetic resonance imaging data

MR images were acquired between the years 2008 and 2019. Images were downloaded from the Picture archiving and communication system (PACS) server, de-identified using in-house software and subsequently converted to the Neuroimaging Informatics Technology Initiative (NIfTI) format. All patients underwent a standard institutional imaging protocol for triphasic MR image acquisition. Arterial phase images were used for liver segmentation because most HCC lesions display arterial phase hyperenhancement (APHE), which is reflected in the current LI-RADS criteria [46]. Tumors with APHE exhibit good contrast and high signal-to-noise ratio which facilitates tumor delineation. Late arterial-phase T1-weighted breath-hold sequences were acquired 12–18 seconds (s) post-contrast injection with several gadolinium-based contrast agents. Images were acquired on a variety of scanners with different field strengths (1.16T, 1.5T, and 3T). Full details of the imaging parameters can be found in the (S1 Table). Briefly, the median repetition time (TR) and median echo time (TE) were 4.39 ms and 2 ms, respectively. The median slice thickness was 3 mm, the median bandwidth 445 Hz, and

the image matrix ranged from 1406×138 to 3206×247. All liver segmentations were done by a medical student (M.G., over 2.5 years of image analysis training) under the supervision of a board-certified abdominal radiologist (M.S., 10 years of experience) using 3D Slicer (v4.10.2) [47].

## Data partition

Early and intermediate BCLC stage (i.e., BCLC-0, BCLC-A, BCLC-B) patients were randomly split into training, validation, and testing sets containing 70%, 15%, and 15% of the subjects, respectively. Due to the relatively lower number of data samples of late BCLC stages (i.e., BCLC-C, BCLC-D), these subjects were split among the training and testing sets to each contain 50% of the subjects, respectively. The sampled subjects from the set of training data were then used to create two equally sized training subsets.

## Model development

Two deep neural networks were trained in a supervised manner to automatically segment the liver from 3D arterial-phase MR images. Both models have an identical fully-convolutional encoder-decoder architecture [48] based on the U-net [49] that includes residual units [50] and uses 3D convolution operations (see Sec. S1 File for details). The only difference between the two algorithms were the datasets used for training, which were composed from different combinations of BCLC stages. The first model, "Early-Intermediate-Stage-Net" (EIS-Net), was trained on early and intermediate BCLC stages. The second model, "All-Stage-Net" (AS-Net), was trained using a dataset comprised of all five BCLC stages. Both models used the same validation set and were tested on the same test set. The manual liver segmentations were used as ground-truth.

The input MR images were standardized to have isotropic voxel spacing of 2mm$^3$ and intensities were scaled so that the 25$^{th}$ and 75$^{th}$ percentile ranged between -0.5 and +0.5 [51]. For model training, random 3D image patches (64×64×32 voxels) were extracted in a 3:1 ratio centered on the liver mask compared to the background image to focus model training on the liver. Both models were trained over 2000 epochs using mini-batches of 64 patches and the Dice similarity loss function [52] using the Adam optimizer [53] with a fixed learning rate of 0.0001. Dice loss was optimized as this metric represents evaluation of the segmentation task at hand. The framework for model training and evaluation is depicted in Fig 1.

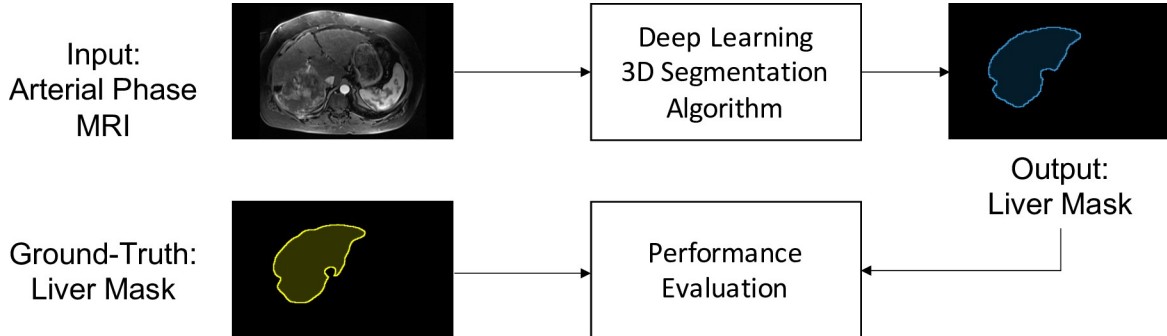

**Fig 1. Overview of the training and evaluation framework for the automated 3D liver segmentation method.** Training input consists of 3D arterial-phase magnetic resonance image (MRI) volumes with corresponding manually annotated ground-truth liver segmentation masks. To evaluate model performance in an independent test set, the output liver segmentations were compared to annotated ground-truth.

Models were implemented in Python (v3.7) using PyTorch (v1.5.1) and the open-source Medical Open Network for AI (MONAI) (v0.3.0) framework. Model training and evaluation was performed on a Linux workstation using an NVIDIA RTX 2080 Ti GPU. All code is publicly available under https://github.com/OnofreyLab/liver-segm.

## Model evaluation and statistical analysis

The two algorithms' 3D liver segmentations were assessed qualitatively and compared quantitatively against the manual segmentations. To quantify segmentation performance, the Dice Similarity Coefficient (DSC) was calculated to measure overlap with the ground-truth. The worst-case segmentation surface accuracy of the algorithms' liver segmentation to the ground-truth was evaluated by means of a Modified Hausdorff Distance (MHD). Here, the MHD was defined as the 95$^{th}$ percentile of the original Hausdorff Distance (HD) since HD was shown to be sensitive to outliers [54]. To assess average segmentation surface accuracy, the Mean Absolute Distance (MAD) of the output liver segmentation mask to the ground-truth was calculated. The units for MHD and MAD were calculated in voxels (for images with 2mm$^3$ voxel spacing). Equations for the segmentation metrics can be found in the S1 File.

Descriptive statistics were calculated using the Python library SciPy (v1.5.2) and were reported as absolute and relative frequencies (n and %) for categorical variables, mean and standard deviation (SD) for normally distributed variables, or median and interquartile range (IQR) for not normally distributed variables. A Wilcoxon signed-rank test was used for statistical pairwise comparisons between the algorithms and a p-value <0.05 was considered significant.

## Compliance with ethical standards

This HIPAA-compliant retrospective, single-institution study was conducted in accordance with the Declaration of Helsinki, and approval was granted by the Institutional Review Board of the Yale University School of Medicine with waiver of informed consent.

## Results

### Study population

From an institutional database of 629 HCC subjects, 219 subjects met the defined inclusion criteria. Population sample statistics are summarized in Table 1 and MR imaging parameters are summarized in the (S1 Table). Briefly, the study population comprised 170 male (77.6%) and 49 female (22.4%) subjects with an age distribution of 62.8±9.1 (mean±SD) years with treatment-naïve HCC. Thirty (13.7%) patients were staged as BCLC-0, 122 (55.7%) as BCLC-A, 32 (14.6%) as BCLC-B, 15 (6.8%) as BCLC-C, and 20 (9.1%) as BCLC-D.

### Data split

Each of the two training sets consisted of 129 patients: For the "Early-Intermediate-Stage-Net" (EIS-Net), the training set comprised of 21 (16.2%) BCLC-0, 86 (66.6%) BCLC-A, and 22 (17.1%) BCLC-B patients; the training set for the "All-Stage-Net" (AS-Net) comprised of 19 (14.7%) BCLC-0, 74 (57.3%) BCLC-A, 18 (14.0%) BCLC-B, 8 (6.2%) BCLC-C, and 10 (7.7%) BCLC-D patients. Both algorithms shared the same validation set comprised of 28 patients with the following BCLC stages: Four (14.3%) BCLC-0, 19 (67.8%) BCLC-A and 5 (17.9%) BCLC-B patients and were evaluated on the same test set consisting of 44 patients comprised by the following cancer stages: 5 (11.4%) BCLC-0, 17 (38.6%) BCLC-A, 5 (11.4%) BCLC-B, 7

**Table 1. Demographic, radiological, and cancer staging sample statistics of the training, validation, and testing cohorts from 219 HCC patients included in this study.**

| Parameter | | Overall | Training | | | Validation | Testing |
|---|---|---|---|---|---|---|---|
| | | | Training Pool | EIS-Net | AS-Net | | |
| **n** | | 219 | 147 | 129 | 129 | 28 | 44 |
| **Demographics** | | | | | | | |
| Age, mean (SD) | | 62.8 | 63.2 | 62.5 | 63.1 | 61.4 | 62.3 |
| | | (9.1) | (8.6) | (8.5) | (8.6) | (11.4) | (9.2) |
| Gender | F | 49 | 38 | 37 | 34 | 4 | 7 |
| | | (22.4) | (25.9) | (28.7) | (26.4) | (14.3) | (15.9) |
| | M | 170 | 109 | 92 | 95 | 24 | 37 |
| | | (77.6) | (74.1) | (71.3) | (73.6) | (85.7) | (84.1) |
| Ethnicity | Asian | 7 | 5 | 5 | 4 | 1 | 1 |
| | | (3.2) | (3.4) | (3.9) | (3.1) | (3.6) | (2.3) |
| | Black, Non-Hispanic | 28 | 20 | 19 | 17 | 0 | 8 |
| | | (12.8) | (13.6) | (14.7) | (13.2) | | (18.2) |
| | Hispanic | 27 | 15 | 14 | 13 | 7 | 5 |
| | | (12.3) | (10.2) | (10.9) | (10.1) | (25.0) | (11.4) |
| | Other/Unknown | 4 | 1 | 1 | 1 | 2 | 1 |
| | | (1.8) | (0.7) | (0.8) | (0.8) | (7.1) | (2.3) |
| | White, Non-Hispanic | 153 | 106 | 90 | 94 | 18 | 29 |
| | | (69.9) | (72.1) | (69.8) | (72.9) | (64.3) | (65.9) |
| Cirrhosis | absent | 12 | 7 | 5 | 7 | 1 | 4 |
| | | (5.5) | (4.8) | (3.9) | (5.4) | (3.6) | (9.1) |
| | present | 207 | 140 | 124 | 122 | 27 | 40 |
| | | (94.5) | (95.2) | (96.1) | (94.6) | (96.4) | (90.9) |
| Etiology | HCV | 125 | 89 | 83 | 79 | 13 | 23 |
| | | (57.1) | (60.5) | (64.3) | (61.2) | (46.4) | (52.3) |
| | HBV | 14 | 11 | 9 | 10 | 1 | 2 |
| | | (6.4) | (7.5) | (7.0) | (7.8) | (3.6) | (4.5) |
| | Alcohol | 60 | 42 | 36 | 33 | 10 | 8 |
| | | (27.4) | (28.6) | (27.9) | (25.6) | (35.7) | (18.2) |
| | NASH | 30 | 17 | 13 | 14 | 6 | 7 |
| | | (13.7) | (11.6) | (10.1) | (10.9) | (21.4) | (15.9) |
| | Autoimmune | 4 | 2 | 2 | 1 | 0 | 2 |
| | | (1.8) | (1.4) | (1.6) | (0.8) | | (4.5) |
| | Cryptogenic | 3 | 3 | 3 | 3 | 0 | 0 |
| | | (1.4) | (2.0) | (2.3) | (2.3) | | |
| | not available | 6 | 3 | 0 | 3 | 0 | 3 |
| | | (2.7) | (2.0) | | (2.3) | | (6.8) |
| **Radiological data** | | | | | | | |
| Liver volume (ccm), median [Q1, Q3] | | 1596.6 | 1599.3 | 1599.3 | 1561.6 | 1753.8 | 1534.0 |
| | | [1271.3, 2054.8] | [1259.5, 2031.5] | [1229.0, 1989.3] | [1228.5, 2053.1] | [1403.6, 2554.7] | [1304.3, 1953.1] |

(*Continued*)

**Table 1.** (Continued)

| Parameter | | Overall | Training | | | Validation | Testing |
|---|---|---|---|---|---|---|---|
| | | | Training Pool | EIS-Net | AS-Net | | |
| Number of lesions | 1 | 147 | 98 | 87 | 87 | 17 | 32 |
| | | (67.1) | (66.7) | (67.4) | (67.4) | (60.7) | (72.7) |
| | 2 | 36 | 27 | 24 | 24 | 6 | 3 |
| | | (16.4) | (18.4) | (18.6) | (18.6) | (21.4) | (6.8) |
| | 3 | 18 | 12 | 11 | 9 | 3 | 3 |
| | | (8.2) | (8.2) | (8.5) | (7.0) | (10.7) | (6.8) |
| | >3 | 18 | 10 | 7 | 9 | 2 | 6 |
| | | (8.2) | (6.8) | (5.4) | (7.0) | (7.1) | (13.6) |
| Maximum tumor diameter (cm), median [Q1, Q3] | | 2.6 | 2.6 | 2.5 | 2.6 | 2.3 | 3.5 |
| | | [2.0, 4.0] | [2.0, 3.7] | [1.9,3.4] | [2.0, 3.7] | [2.1, 3.1] | [2.4, 5.0] |
| Cumulative tumor diameter (cm), median [Q1, Q3] | | 3.0 | 3.0 | 2.9 | 3.2 | 3.0 | 3.2 |
| | | [2.0, 5.2] | [2.0, 5.0] | [2.0, 4.3] | [2.0, 5.0] | [2.1, 4.5] | [2.4, 6.1] |
| Liver lobe | bilobar | 42 | 31 | 25 | 25 | 4 | 7 |
| | | (19.2) | (21.1) | (19.4) | (19.4) | (14.3) | (15.9) |
| | left | 42 | 23 | 20 | 20 | 8 | 11 |
| | | (19.2) | (15.6) | (15.5) | (15.5) | (28.6) | (25.0) |
| | right | 135 | 93 | 84 | 84 | 16 | 26 |
| | | (61.6) | (63.3) | (65.1) | (65.1) | (57.1) | (59.1) |
| Disease involves >50% of the liver parenchyma | no | 199 | 136 | 123 | 119 | 27 | 36 |
| | | (90.9) | (92.5) | (95.3) | (92.2) | (96.4) | (81.8) |
| | yes | 20 | 11 | 6 | 10 | 1 | 8 |
| | | (9.1) | (7.5) | (4.7) | (7.8) | (3.6) | (18.2) |
| Ascites on imaging | absent | 167 | 111 | 100 | 100 | 21 | 35 |
| | | (76.3) | (75.5) | (77.5) | (77.5) | (75.0) | (79.5) |
| | moderate | 17 | 12 | 7 | 9 | 1 | 4 |
| | | (7.8) | (8.2) | (5.4) | (7.0) | (3.6) | (9.1) |
| | slight | 35 | 24 | 22 | 20 | 6 | 5 |
| | | (16.0) | (16.3) | (17.1) | (15.5) | (21.4) | (11.4) |
| Portal hypertension on imaging | absent | 102 | 69 | 61 | 63 | 13 | 20 |
| | | (46.6) | (46.9) | (47.3) | (48.8) | (46.4) | (45.5) |
| | present | 117 | 78 | 68 | 66 | 15 | 24 |
| | | (53.4) | (53.1) | (52.7) | (51.2) | (53.6) | (54.5) |
| Portal vein thrombosis | absent | 205 | 139 | 127 | 121 | 28 | 38 |
| | | (93.6) | (94.6) | (98.4) | (93.8) | (100.0) | (86.4) |
| | present | 14 | 8 | 2 | 8 | 0 | 6 |
| | | (6.4) | (5.4) | (1.6) | (6.2) | | (13.6) |
| Tumor thrombus | absent | 206 | 140 | 129 | 122 | 28 | 38 |
| | | (94.1) | (95.2) | (100.0) | (94.6) | (100.0) | (86.4) |
| | present | 13 | 7 | 0 | 7 | 0 | 6 |
| | | (5.9) | (4.8) | | (5.4) | | (13.6) |
| Infiltrative | no | 208 | 141 | 129 | 123 | 28 | 39 |
| | | (95.0) | (95.9) | (100.0) | (95.3) | (100.0) | (88.6) |
| | yes | 11 | 6 | 0 | 6 | 0 | 5 |
| | | (5.0) | (4.1) | | (4.7) | | (11.4) |
| **Staging system** | | | | | | | |

(Continued)

**Table 1.** (Continued)

| Parameter | | Overall | Training | | | Validation | Testing |
|---|---|---|---|---|---|---|---|
| | | | Training Pool | EIS-Net | AS-Net | | |
| Child-Pugh Class | A | 140 | 92 | 86 | 83 | 18 | 30 |
| | | (63.9) | (62.6) | (66.7) | (64.3) | (64.3) | (68.2) |
| | B | 64 | 48 | 43 | 39 | 10 | 6 |
| | | (29.2) | (32.7) | (33.3) | (30.2) | (35.7) | (13.6) |
| | C | 15 | 7 | 0 | 7 | 0 | 8 |
| | | (6.8) | (4.8) | | (5.4) | | (18.2) |
| ECOG performance status | 0 | 168 | 111 | 107 | 96 | 26 | 31 |
| | | (76.7) | (75.5) | (82.9) | (74.4) | (92.9) | (70.5) |
| | 1 | 34 | 26 | 21 | 23 | 2 | 6 |
| | | (15.5) | (17.7) | (16.3) | (17.8) | (7.1) | (13.6) |
| | 2 | 8 | 6 | 1 | 6 | 0 | 2 |
| | | (3.7) | (4.1) | (0.8) | (4.7) | | (4.5) |
| | 3 | 3 | 1 | 0 | 1 | 0 | 2 |
| | | (1.4) | (0.7) | | (0.8) | | (4.5) |
| | 4 | 6 | 3 | 0 | 3 | 0 | 3 |
| | | (2.7) | (2.0) | | (2.3) | | (6.8) |
| BCLC Stage | 0 | 30 | 21 | 21 | 19 | 4 | 5 |
| | | (13.7) | (14.3) | (16.3) | (14.7) | (14.3) | (11.4) |
| | A | 122 | 86 | 86 | 74 | 19 | 17 |
| | | (55.7) | (58.5) | (66.7) | (57.4) | (67.9) | (38.6) |
| | B | 32 | 22 | 22 | 18 | 5 | 5 |
| | | (14.6) | (15.0) | (17.1) | (14.0) | (17.9) | (11.4) |
| | C | 15 | 8 | 0 | 8 | 0 | 7 |
| | | (6.8) | (5.4) | | (6.2) | | (15.9) |
| | D | 20 | 10 | 0 | 10 | 0 | 10 |
| | | (9.1) | (6.8) | | (7.8) | | (22.7) |

Numbers in parentheses are percentages if not indicated otherwise. EIS-Net = Early-Intermediate-Stage-Net, AS-Net = All-Stage-Net, HBV = Hepatitis B Virus, HCV = Hepatitis C Virus, ECOG = Eastern Cooperative Oncology Group, BCLC = Barcelona Clinic Liver Cancer.

(15.9%) BCLC-C, and 10 (22.7%) BCLC-D patients. Full details on sampling of the data sets can be found in the flowchart in Fig 2.

## Model performance

Both the EIS- and AIS-net models were trained for 2000 epochs, at which time the loss function of the two models converged on both the training and validation datasets. The DSC (mean±SD) performance on the training datasets were 0.952±0.042 and 0.951±0.035 and on the validation dataset 0.928 ±0.093 and 0.928±0.093 for the EIS-Net and AS-Net, respectively. Segmentation of the validation and test set data was performed on the whole image using a large patch (224x224x128) in order to avoid stitching artifacts from smaller, overlapping patches. Segmentation times (median [IQR]) for both the EIS- and AS-Net were 0.73 [0.33] seconds and 0.70 [0.27] seconds on the validation and test set, respectively.

Qualitative assessment of the algorithms' segmentation outputs on the test set across different BCLC stages showed that both the EIS-Net and the AS-Net performed well on early and intermediate BCLC stages (i.e., BCLC-0, BCLC-A, BCLC-B). However, the AS-Net

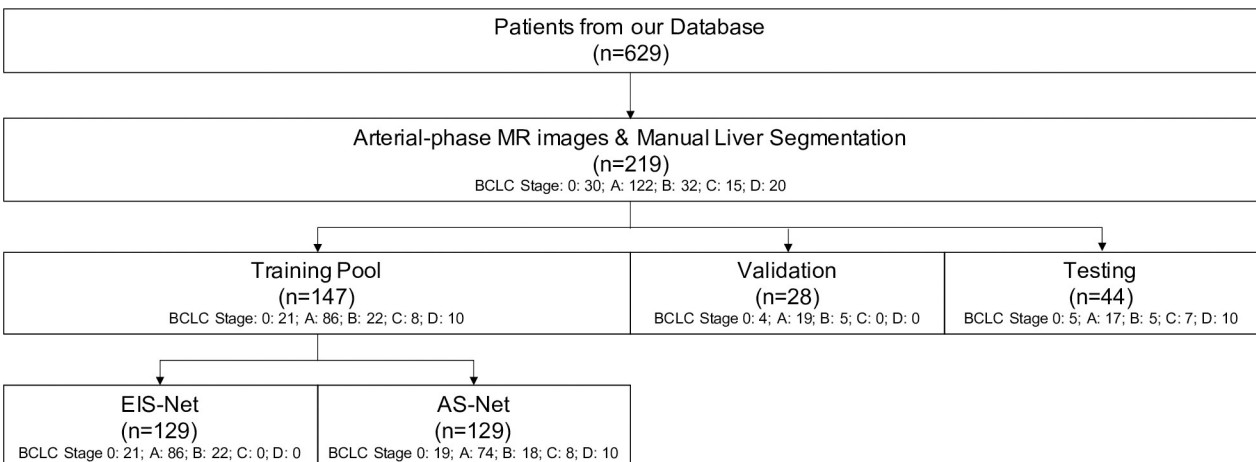

**Fig 2. Inclusion and exclusion criteria, and partitioning of the dataset for model training and evaluation.** From an institutional database, 219 HCC patients that had arterial-phase MR images and a manual liver segmentation available for processing were included. Subjects from each BCLC stage were then allocated to the test set and patients were selected for shared validation and testing sets. From the overall training pool, subjects were sampled to create two training data subsets for the Early-Intermediate-Stage-Net (EIS-Net) and the All-Stage-Net (AS-Net).

outperformed the EIS-Net on more advanced stages (i.e., BCLC-C and BCLC-D). Examples of representative liver segmentations across BCLC stages are shown in Fig 3.

Detailed assessments of the segmentation results showed that the EIS-Net failed on some advanced BCLC cancer stages with big HCC tumors, where large areas of hypointense necrotic

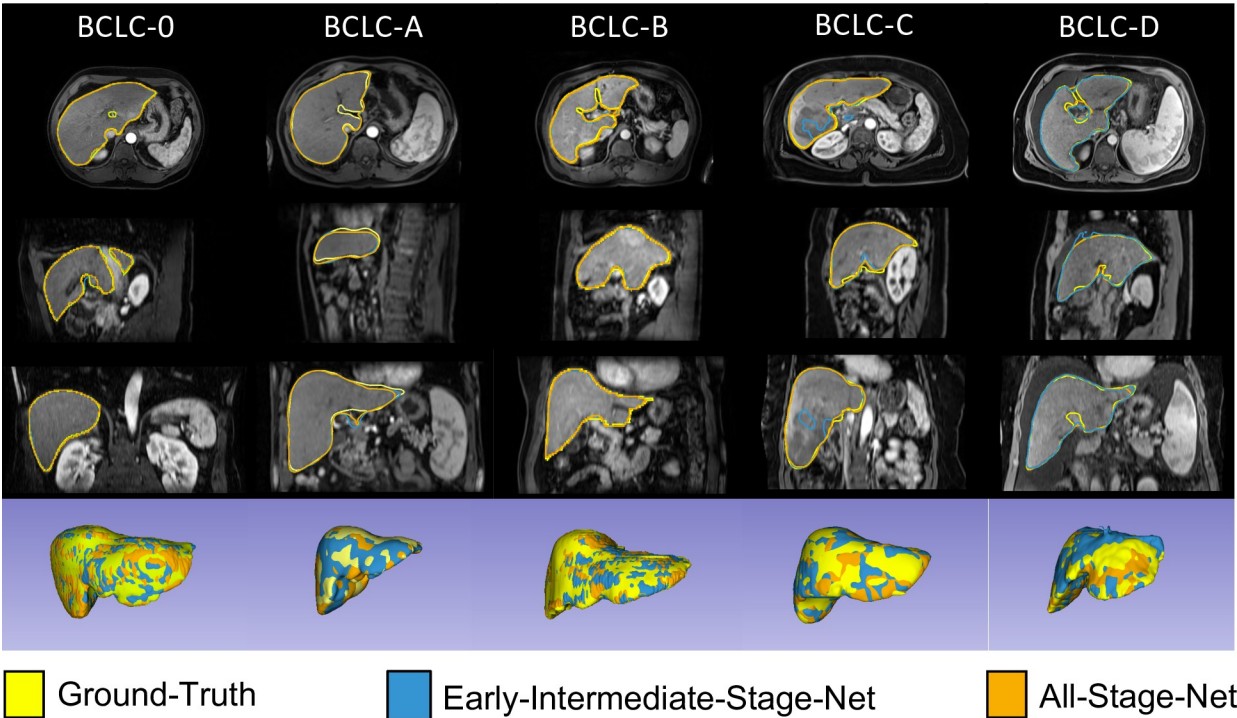

**Fig 3. Example liver segmentations results across Barcelona Clinic Liver Cancer (BCLC) stages.** Rows from top to bottom show axial, sagittal and coronal arterial-phase magnetic resonance images of different subjects across BCLC stages (from left to right). The last row displays the liver segmentations as 3D renderings. The liver segmentation masks of the Early-Intermediate-Stage-Net (blue) and All-Stage-Net (orange), as well as the ground-truth (yellow) are overlaid on the images. While the Early-Intermediate-Stage-Net was trained only on patients with BCLC stages 0, A and B, the All-Stage-Net was trained on a training set spanning all BCLC cancer stages.

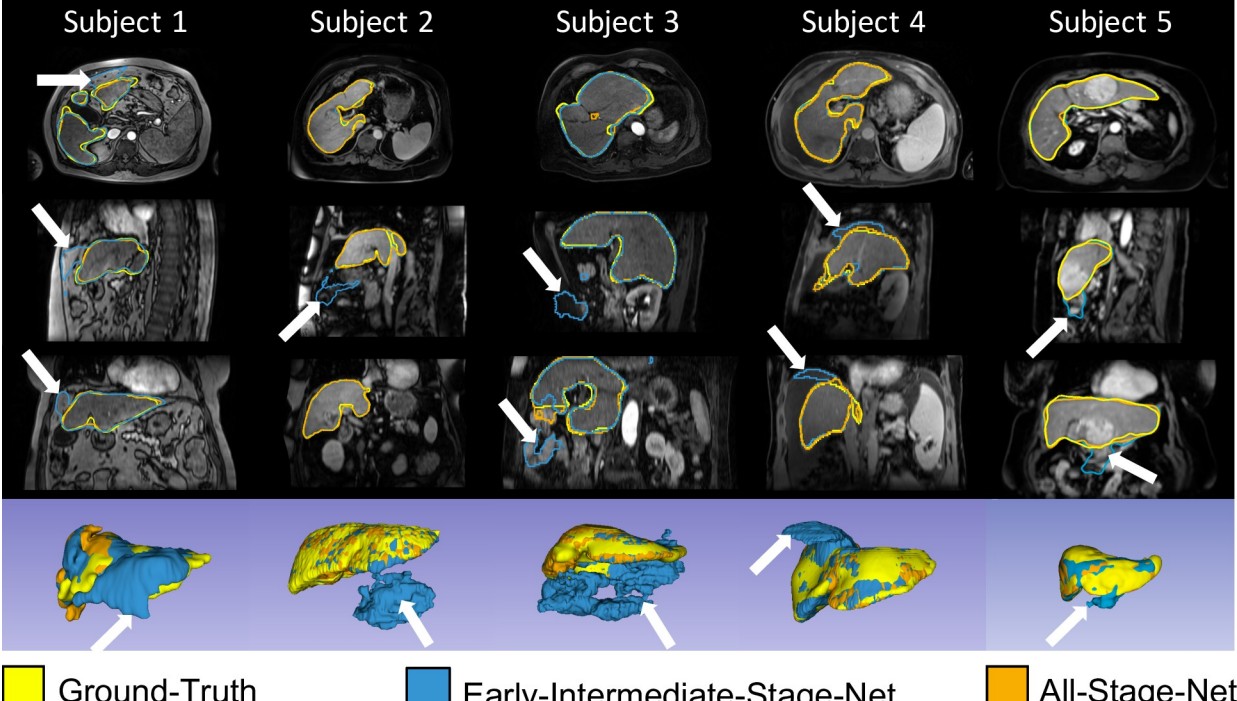

**Fig 4. Examples of the superior liver segmentation performance of the All-Stage-Net over the Early-Intermediate-Stage-Net.** Columns show results from five different subjects. Rows from top to bottom show axial, sagittal and coronal arterial-phase magnetic resonance images on which the All-Stage-Net (overlaid in orange) outperformed the Early-Intermediate-Stage-Net (overlaid in blue). Expert ground-truth liver segmentations are overlaid in yellow. The last row displays the liver segmentations as 3D renderings. White arrows point on areas of liver segmentation failure of the Early-Intermediate-Stage-Net. While the Early-Intermediate-Stage-Net was trained only on patients with Barcelona Clinic Liver Cancer (BCLC) stages 0, A and B, the All-Stage-Net was trained on a training set spanning all BCLC cancer stages.

tumor tissue were not classified as liver tissue. The AS-Net, by contrast, correctly classified those regions as liver tissue. In other cases, the EIS-Net incorrectly classified structures around the liver, such as parts of the small intestine or colon, as part of the liver, while the AS-Net correctly delineated the anatomical liver contour in those scans. Furthermore, in some patients, regions of large ascites surrounding the liver were classified as liver parenchyma by the EIS-Net, leading to large over-segmentation of the liver, whereas the AS-Net did not consider these areas to be part of the liver. Representative examples of better liver segmentation results of the AS-Net against the EIS-Net are shown in Fig 4.

Quantitative analysis of the segmentation to the expert ground-truth showed (mean±SD) Dice Similarity Coefficients (DSC) for liver segmentations compared with manual segmentations of 0.946±0.032 and 0.954±0.018 for the EIS-Net and the AS-Net, respectively (p<0.0001). The Modified Hausdorff Distance (MHD) (mean±SD), measuring the closeness of the algorithms' liver segmentation to the manual ground-truth, were 5.812±8.822 and 3.500 ±4.033 for the EIS-Net and AS-Net, respectively (p = 0.005). The Mean Absolute Distance (MAD) (mean±SD) for the liver segmentations compared with the expert segmentations were 1.243±1.901 for the EIS-Net and 0.750±0.370 for the AS-Net (p = 0.005). Further radiological assessment showed that a DSC of 0.95 between the ground-truth and the algorithms' liver segmentation correlated well with the ground-truth.

When the models' liver segmentation performances were compared across different BCLC stages, they did not differ significantly for the early and intermediate BCLC stages (DSC: p = 0.107, MHD: p = 0.413, MAD: p = 0.428) between both liver segmentation models.

**Table 2. Liver segmentation performance (Dice Similarity Coefficient (DSC), Modified Hausdorff Distance (MHD), and Mean Absolute Distance (MAD)) of the EIS-Net and AS-Net methods compared to manual ground-truth across different Barcelona Clinic Liver Cancer (BCLC) cancer stages.**

| | | EIS-Net | | | AS-Net | | | |
| --- | --- | --- | --- | --- | --- | --- | --- | --- |
| | Count | Mean | SD | Median | Mean | SD | Median | p-Value |
| **DSC** Overall | 44 | 0.946 | 0.032 | 0.957 | 0.954 | 0.018 | 0.960 | <0.001 * |
| Early & intermediate stages | 27 | 0.949 | 0.027 | 0.959 | 0.952 | 0.021 | 0.960 | 0.107 |
| BCLC-0 | 5 | 0.959 | 0.011 | 0.959 | 0.961 | 0.010 | 0.962 | 0.312 |
| BCLC-A | 17 | 0.940 | 0.030 | 0.948 | 0.944 | 0.022 | 0.949 | 0.132 |
| BCLC-B | 5 | 0.970 | 0.005 | 0.970 | 0.970 | 0.005 | 0.969 | 0.438 |
| Advanced stages | 17 | 0.941 | 0.038 | 0.954 | 0.956 | 0.014 | 0.962 | <0.001 * |
| BCLC-C | 7 | 0.931 | 0.053 | 0.952 | 0.954 | 0.007 | 0.952 | 0.016 * |
| BCLC-D | 10 | 0.949 | 0.024 | 0.962 | 0.958 | 0.017 | 0.965 | 0.020 * |
| **MHD (in voxels)** Overall | 44 | 5.812 | 8.822 | 2.236 | 3.500 | 4.033 | 2.236 | 0.005 * |
| Early & intermediate stages | 27 | 4.076 | 5.217 | 2.236 | 3.759 | 4.815 | 2.236 | 0.413 |
| BCLC-0 | 5 | 2.232 | 0.159 | 2.236 | 2.213 | 0.359 | 2.000 | 1.000 |
| BCLC-A | 17 | 5.266 | 6.329 | 2.236 | 4.758 | 5.887 | 2.236 | 0.359 |
| BCLC-B | 5 | 1.878 | 0.585 | 1.732 | 1.912 | 0.656 | 1.732 | 0.317 |
| Advanced stages | 17 | 8.570 | 12.320 | 3.464 | 3.089 | 2.398 | 2.236 | 0.003 * |
| BCLC-C | 7 | 8.928 | 14.142 | 3.742 | 3.117 | 0.914 | 3.000 | 0.249 |
| BCLC-D | 10 | 8.319 | 11.675 | 2.532 | 3.069 | 3.109 | 2.236 | 0.012 * |
| **MAD (in voxels)** Overall | 44 | 1.243 | 1.901 | 0.698 | 0.750 | 0.370 | 0.626 | 0.005 * |
| Early & intermediate stages | 27 | 0.856 | 0.556 | 0.678 | 0.814 | 0.449 | 0.699 | 0.428 |
| BCLC-0 | 5 | 0.635 | 0.076 | 0.642 | 0.632 | 0.085 | 0.607 | 0.625 |
| BCLC-A | 17 | 1.002 | 0.660 | 0.705 | 0.943 | 0.523 | 0.759 | 0.782 |
| BCLC-B | 5 | 0.579 | 0.117 | 0.529 | 0.561 | 0.126 | 0.519 | 0.625 |
| Advanced stages | 17 | 1.858 | 2.925 | 0.745 | 0.648 | 0.152 | 0.617 | <0.001 * |
| BCLC-C | 7 | 2.341 | 3.945 | 0.849 | 0.719 | 0.128 | 0.658 | 0.031 * |
| BCLC-D | 10 | 1.520 | 2.128 | 0.657 | 0.598 | 0.153 | 0.575 | 0.014 * |

Quantitative analysis of liver segmentation performances of the Early-Intermediate-Stage-Net (EIS-Net) and All-Stage-Net (AS-Net) compared against the experts' manual segmentations by means of the Dice Similarity Coefficient (DSC), Modified Hausdorff Distance (MHD), and Mean Absolute Distance (MAD). A Wilcoxon signed-rank test was used for pairwise comparisons between the liver segmentation algorithms and a p-value <0.05 was considered statistically significant (denoted with *).

However, the AS-Net performed significantly better on advanced HCC stages (DSC: p<0.0001, MHD: p = 0.003, MAD: p<0.0001). Pairwise comparisons between the EIS-Net and AS-Net for each BCLC stage are shown in Table 2. Boxplots in Fig 5 show that the AS-Net had lower performance variance, better mean performance, fewer outliers and better worst-case performance than the EIS-Net on all BCLC stages across all quantitative segmentation metrics (DSC, MHD, MAD), indicating a more consistent and robust segmentation performance.

In livers where HCC involved <50% of the parenchyma, the AS-Net outperformed the EIS-Net significantly with all performance measures (DSC: p = 0.005, MHD: p = 0.007, MAD: p = 0.046). In livers where ≥50% of the parenchyma was involved by tumor tissue, the AS-Net had significantly better results when the performances were compared by the DSC and MAD (p = 0.023 and p = 0.039, respectively). However, no statistical significance was found between the two algorithms for the MHD (p = 0.225).

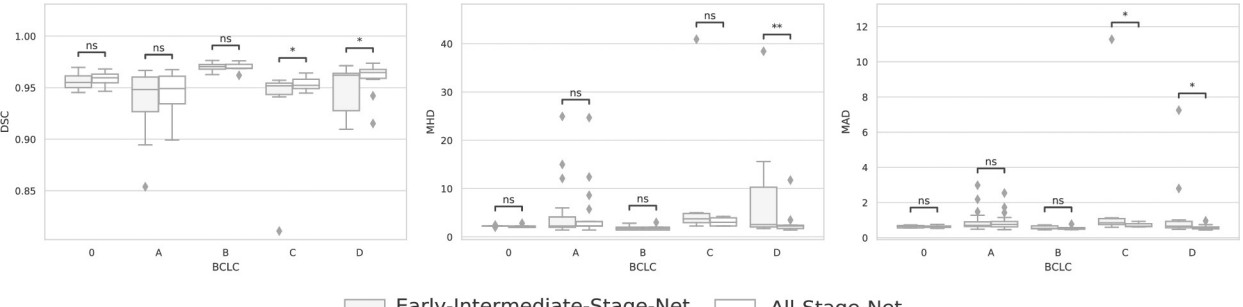

**Fig 5. Liver segmentation method performance across different Barcelona Clinic Liver Cancer (BCLC) cancer stages.** The automatic liver segmentations of the Early-Intermediate-Stage-Net (EIS-Net) and All-Stage-Net (AS-Net) were compared quantitatively against the experts' manual segmentations by means of the Dice Similarity Coefficient (DSC), Modified Hausdorff Distance (MHD), and Mean Absolute Distance (MAD). AS-Net showed better mean performance, fewer outliers and better worst-case performance across all segmentation metrics indicating a more robust segmentation performance. A Wilcoxon signed-rank test was used for pairwise comparisons between the liver segmentation algorithms and a p-value <0.05 was considered statistically significant (denoted with *, ns denotes no significant differences).

When compared specifically for the extent of cumulative tumor diameter, the AS-Net and EIS-Net did not yield statistically significantly different results for tumors <3cm (DSC: p = 0.090, MHD: p = 0.385, MAD: p = 0.142). However, the AS-Net showed significantly better results than the EIS-Net for tumors ≥3cm (DSC: p = 0.002, MHD: p = 0.003, MAD: p = 0.018). Comprehensive pairwise comparisons between the two segmentation models for a range of different patient features can be found in the (S2–S4 Tables).

## Discussion

Accurate and robust whole liver segmentation is key for volumetry assessment to guide treatment decisions when deciding if various treatment options such as liver resection, radioembolization or portal vein embolization are safe [13, 15, 55, 56]. Moreover, liver segmentation is an important pre-processing step for subsequent cancer detection algorithms. Segmentation can be especially challenging in patients with cancer-related tissue changes and liver shape deformity as morphology can be substantially altered. To improve automated segmentation performance on MR images in patients with heterogeneous imaging characteristics across the full spectrum of primary liver cancer, a deep learning algorithm was trained using imaging data spanning the full distribution of BCLC staging.

In this study, we demonstrated that training across the distribution of BCLC stages significantly improved the ability of deep learning liver segmentation algorithms to generalize across cancer stages. Models trained using data across all BCLC stages yielded better and more consistent segmentation performance when compared to models trained only on early and intermediate cancer stages. Both the "Early-Intermediate-Stage-Net" (EIS-Net) and the "All-Stage-Net" (AS-Net) showed good segmentation results on livers with early and intermediate BCLC stages. However, the EIS-Net failed on the segmentation of some advanced BCLC stage patients on which the AS-Net showed robust segmentation results. Overall, training with diverse data reduced the variance in segmentation performance, making deep learning algorithms more robust and able to achieve greater performance consistency across a heterogeneous cohort of imaging data that is typically encountered in clinical practice.

Advanced liver cancer leads to heterogeneous liver tissue and significantly altered liver shapes [22, 23]. Moreover, multifocal and large tumors displaying voluminous areas of contrast-enhancement, tumor necrosis, infiltrative disease, perfusion abnormalities or tumor thrombi considerably change liver tissue morphology on MR images and therefore make it

difficult for deep neural networks to correctly classify those areas as liver tissue. Additionally, the liver contour can be altered by a more cirrhotic configuration displayed as a more nodular surface, and with progressing liver failure and the development of portal hypertension, further alterations including large volume ascites [22, 57]. All these factors substantially change the liver morphology on MR images and make the segmentation task challenging.

We hypothesized that the AS-Net showed better performance on advanced BCLC stage patients since it had already seen much bigger tumors, heterogeneous liver tissue, and severe ascites in its training data. Interestingly, the AS-Net did not perform worse on earlier BCLC stages, even with fewer training data of those stages. Moreover, the diversity in the AS-Net's training data helped the model generalize better on various HCC stages and showed less variance across all performance measures, indicating that the heterogeneity of cancer stages in the training data also helped to improve consistency by reducing distributional shift between the training and testing data [44]. The model also had better worst-case performance, indicating that the diversity of BCLC stages lead to more robust segmentation performance.

Many current state-of-the-art deep learning segmentation algorithms use encoder-decoder network architectures, and many practical improvements in segmentation performance can be realized through innovations in pre-processing [51], data augmentation and loss functions [29]. Previous liver segmentation studies have used the U-net architecture [19, 29, 32, 36, 37, 39] or its variants [34, 38]. The method of Bousabarah et al. [19] trained on 121 triphasic MR scans and tested on a set of 26 patients yielded a mean DSC of 0.91 (±0.01). The proposed fully convolutional neural network of Zeng et al. [34] used T2-weighted MR images and showed a DSC (mean±SD) of 0.952±0.01 on 51 validation patients. Wang et al.'s 2D U-net CNN for liver segmentation yielded a mean DSC of 0.95±0.03 with their method trained using 330 MRI and CT scans and tested on 100 T1-weighted MR images [32]. While our study's goal was to determine the relative effect of different training data cohorts on segmentation model performance and not to focus on obtaining peak segmentation performance by exhaustively optimizing the network architecture, both of our models demonstrated segmentation performance comparable to that of previously published studies. Further performance gains may be realized with additional network tuning and model training strategies, and future work will involve accounting for distributional shifts during the model training process [58].

Our study has several limitations. First, the data for the staging of the patients of our database was collected retrospectively from the electronic health record of the hospital, and most patients in our database are distributed among earlier BCLC stages. Nevertheless, this distribution accurately reflects the clinical population at this site as most patients who undergo contrast-enhanced MR image acquisitions that require breath-holding are distributed across earlier BCLC stages and patients with more advanced disease and resultant poor performance status are unable to successfully complete the necessary instructions required for adequate MR image acquisition. Additionally, our data was limited to treatment-naïve HCC patients and did not include patients with other types of hepatic pathologies. Therefore, we were not able to investigate how treatment-associated changes of the liver parenchyma would affect the models' segmentation performance. Future work will assess the performance of the algorithm on patients who underwent treatment and include a prospective evaluation of AS-Net using data from multiple sites, as well as verifying that our results hold across different network architectures.

## Conclusion

In this paper, we demonstrate the training and validation of a fully automated 3D liver segmentation method using deep learning across the full spectrum of BCLC cancer stages. Our

results show that diversity in the training data across all BCLC stages significantly improves the performance of robust whole liver MRI segmentation algorithms compared to the same algorithm trained with images representative of a limited subset of BCLC stages. To avoid problems caused by distributional shift and to ensure robust segmentation performance that is independent of liver shape deformation and tumor burden and generalizable across BCLC cancer stages, it is critical to train deep learning models on heterogeneous imaging data spanning all cancer stages and a diverse spectrum of diagnostic features. Moreover, we demonstrate the importance of model validation on datasets that are composed of a spectrum of cancer stages that exhibit heterogeneous diagnostic findings encountered in clinical practice.

## Supporting information

**S1 Table. Magnetic resonance imaging parameters.** Magnetic resonance imaging parameters of the training, validation, and testing cohorts from 219 HCC patients included in this study.
(DOCX)

**S2 Table. Dice Similarity Coefficient (DSC) results.** Dice Similarity Coefficient (DSC) results for the Early-Intermediate-Stage-Net (EIS-Net) and All-Stage-Net (AS-Net) compared against the experts' manual segmentations.
(DOCX)

**S3 Table. Modified Hausdorff Distance (MHD) results.** Modified Hausdorff Distance (MHD) (in voxels) results for the Early-Intermediate-Stage-Net (EIS-Net) and All-Stage-Net (AS-Net) compared against the experts' manual segmentations.
(DOCX)

**S4 Table. Mean Absolute Distance (MAD) results.** Mean Absolute Distance (MAD) (in voxels) results for the Early-Intermediate-Stage-Net (EIS-Net) and All-Stage-Net (AS-Net) compared against the experts' manual segmentations.
(DOCX)

**S1 File.**
(DOCX)

## Author Contributions

**Conceptualization:** Moritz Gross, Julius Chapiro, John A. Onofrey.

**Data curation:** Moritz Gross, Ariel Jaffe, Ahmet S. Kucukkaya, Simon Iseke, John A. Onofrey.

**Formal analysis:** Moritz Gross, John A. Onofrey.

**Funding acquisition:** John A. Onofrey.

**Investigation:** Moritz Gross, John A. Onofrey.

**Methodology:** Moritz Gross, Julius Chapiro, John A. Onofrey.

**Project administration:** Julius Chapiro, John A. Onofrey.

**Resources:** Julius Chapiro, John A. Onofrey.

**Software:** Moritz Gross, John A. Onofrey.

**Supervision:** Michael Spektor, Mario Strazzabosco, Julius Chapiro, John A. Onofrey.

**Validation:** Moritz Gross, Michael Spektor, Ariel Jaffe, Stefan P. Haider, Julius Chapiro, John A. Onofrey.

**Visualization:** Moritz Gross, John A. Onofrey.

**Writing – original draft:** Moritz Gross, John A. Onofrey.

**Writing – review & editing:** Moritz Gross, Michael Spektor, Ariel Jaffe, Ahmet S. Kucukkaya, Simon Iseke, Stefan P. Haider, Mario Strazzabosco, Julius Chapiro, John A. Onofrey.

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
