## [Decision Letter · Decision Letter 0]

13 Jul 2021

PONE-D-21-18482

Improved performance and consistency of deep learning 3D liver segmentation with heterogenous cancer stages in magnetic resonance imaging

PLOS ONE

Dear Dr. Onofrey,

Thank you for submitting your manuscript to PLOS ONE. After careful consideration, we feel that it has merit but does not fully meet PLOS ONE’s publication criteria as it currently stands. Therefore, we invite you to submit a revised version of the manuscript that addresses the points raised during the review process.

Please revise according to reviewers' comments especially the comment to compare the proposed techniques with the state-of-arts to further demonstrate the contributions of the research work.

We look forward to receiving your revised manuscript.

Kind regards,

Yan Chai Hum

Academic Editor

PLOS ONE

Journal Requirements:

[Research reported in this publication was supported by the National Institute of Diabetes and Digestive and Kidney Diseases of the National Institutes of Health under Award Number P30 KD034989 and the National Institutes of Health Grant Award Number DDRCC DK034989-36 for the Clinical Translational Core of the Yale Liver Center. The content is solely the responsibility of the authors and does not necessarily represent the official views of the Nation Institutes of Health. M.G. was supported by a travel grant by the Rolf W. Günther Foundation for Radiological Sciences for travel to Yale University.]

 [J.O. was supported by the National Institute of Diabetes and Digestive and Kidney Diseases of the National Institutes of Health under Award Number P30 KD034989 and M.S. the National Institutes of Health Grant Award Number DDRCC DK034989-36 for the Clinical Translational Core of the Yale Liver Center. The content is solely the responsibility of the authors and does not necessarily represent the official views of the Nation Institutes of Health. M.G. was supported by a travel grant by the Rolf W. Günther Foundation for Radiological Sciences for travel to Yale University.The funders had no role in study design, data collection and analysis, decision to publish, or preparation of the manuscript.]

3. your Data Availability statement, you have not specified where the minimal data set underlying the results described in your manuscript can be found. PLOS defines a study's minimal data set as the underlying data used to reach the conclusions drawn in the manuscript and any additional data required to replicate the reported study findings in their entirety. All PLOS journals require that the minimal data set be made fully available. For more information about our data policy, please see http://journals.plos.org/plosone/s/data-availability.

6. We note that Figures 1, 3 and 4 in your submission contain copyrighted images. All PLOS content is published under the Creative Commons Attribution License (CC BY 4.0), which means that the manuscript, images, and Supporting Information files will be freely available online, and any third party is permitted to access, download, copy, distribute, and use these materials in any way, even commercially, with proper attribution. For more information, see our copyright guidelines: http://journals.plos.org/plosone/s/licenses-and-copyright.

a) You may seek permission from the original copyright holder of Figures 1, 3 and 4 to publish the content specifically under the CC BY 4.0 license. 

Reviewers' comments:

Reviewer's Responses to Questions

**Comments to the Author**

1. Is the manuscript technically sound, and do the data support the conclusions?

Reviewer #1: Yes

Reviewer #2: Partly

2. Has the statistical analysis been performed appropriately and rigorously? 

Reviewer #1: Yes

Reviewer #2: Yes

3. Have the authors made all data underlying the findings in their manuscript fully available?

Reviewer #1: Yes

Reviewer #2: Yes

4. Is the manuscript presented in an intelligible fashion and written in standard English?

Reviewer #1: Yes

Reviewer #2: No

5. Review Comments to the Author

Reviewer #1: Well written article. The presented works are reproducible, proposed methodology writing is clear and easy to follow. The writing content demonstrated a high technically sound piece of research works. Presented outcome is convincing with enough simulated samples.

Reviewer #2: In this research, authors wish to assess the ability of state-of-the-art deep learning 3D liver segmentation algorithms to generalize across different BCLC liver cancer stages. The research is interesting, however, the proposed methodology existed in the literature and thus the novelty of the research is not sufficient to be published in PLOS ONE. Furthermore, the discussions and also the analysis of the manuscript need to be further improve in order to be published in PLOS ONE. So I suggest reject the article and giving the author a chance to submit after doing additional research to improve the contributions of the research work. Authors may also consider the following comments for the revision work:

1. The writing of the manuscript need to be further improve. The sections and sub sections need to be change. E.g. there should be sections like Literature review, Methodology, Results and Discussions etc. There are some crucial sections missing in the manuscript. The flow is a bit difficult to follow.

2. The format of the references and citations need to be improve. For the literature review, authors should refer to more recent research, i.e. year 2019-2020. Currently there is only 1 2020 reference which is not sufficient.

3. Authors need to identify the research gap and highlight the contribution(s) of the research work in the manuscript to shows the novelty of the research.

4. Authors should compare the proposed techniques with the state-of-arts to further prove the contribution(s)/novelty of the research work.

5. More scientific reasoning should be added in the experimental results' explanations.

6. The format of the tables need to be improved. The current one is a bit difficult to read.

6. PLOS authors have the option to publish the peer review history of their article (what does this mean?). If published, this will include your full peer review and any attached files.

Reviewer #1: No

Reviewer #2: No

---

## [Author Response · Author response to Decision Letter 0]

27 Aug 2021

We thank the reviewers and editor for their time spent reading our manuscript and for their thoughtful comments. We have prepared detailed responses to each comment and have incorporated changes into our revised manuscript using Track Changes (with changes highlighted in blue text). We provide here a summary of major changes made to this revised manuscript. Detailed responses to individual comments can be found in our Response to Reviewers.

Summary of Major Changes

The revised version of our manuscript includes the following major changes:

1. In response to Reviewer 2, we have revised the manuscript to highlight the contributions and significance:

 a. We have elaborated on the challenges of distributional shift in machine learning in both the Introduction and the Discussion. Our results demonstrate the problem of distributional shift in liver segmentation and how training across the distribution clinical liver cancer imaging data can make model performance more robust by reducing this shift.

 b. We have provided additional references to recent liver segmentation approaches to make our manuscript more comprehensive in both the Introduction and the Discussion sections.

2. Addressed specific journal requirements for submission that include:

 a. Ensured that the manuscript meets PLOS ONE’s style requirements.

 b. Removal of all funding-related text from the manuscript’s Acknowledgements section.

 c. Updated the Data Availability statement.

 d. Moved the Ethics Statement to appear within the Methods section.

 e. All figures are available under the CC BY 4.0 license.

---

## [Editor Report · Decision Letter 1]

4 Oct 2021

PONE-D-21-18482R1Improved performance and consistency of deep learning 3D liver segmentation with heterogenous cancer stages in magnetic resonance imagingPLOS ONE

Dear Dr. Onofrey,

Thank you for submitting your manuscript to PLOS ONE. After careful consideration, we feel that it has merit but does not fully meet PLOS ONE’s publication criteria as it currently stands. Therefore, we invite you to submit a revised version of the manuscript that addresses the points raised during the review process.

 Please improve each caption of figure and title for table to assure that take-home message was incorporated into the caption or title so that the figure can be standalone without requiring readers to read the text to understand the gist of the figure/table. In other words, highlight the findings of the figure/table using caption/title to facilitate reading. Besides, please assure that the Github link can be accessed,. Lastly, you are advised to revise your conclusion based on the result acquired from the experiment; reiterate and specify the critical findings based on experimental result and then only explain the implication of these findings to the field of study. 

We look forward to receiving your revised manuscript.

Kind regards,

Yan Chai Hum

Academic Editor

PLOS ONE
---

## [Author Response · Author response to Decision Letter 1]

5 Nov 2021

We thank the reviewers and editor for their time spent reading our manuscript and for their thoughtful comments. We have prepared detailed responses to each comment and have incorporated changes into our revised manuscript using Track Changes (with changes highlighted in blue text).

Reviewers' Comments:

1. Please improve each caption of figure and title for table to assure that take-home message was incorporated into the caption or title so that the figure can be standalone without requiring readers to read the text to understand the gist of the figure/table. In other words, highlight the findings of the figure/table using caption/title to facilitate reading.

We have changed the figure captions and table titles so that they contain and highlight the main message and can be understood without reading the text.

2. Besides, please assure that the Github link can be accessed,.

The Github link can now be accessed through the following link: https://github.com/OnofreyLab/liver-segm

3. Lastly, you are advised to revise your conclusion based on the result acquired from the experiment; reiterate and specify the critical findings based on experimental result and then only explain the implication of these findings to the field of study.

We have revised our conclusion and made it more specific to our study and field of research.

---

## [Editor Report · Decision Letter 2]

15 Nov 2021

Improved performance and consistency of deep learning 3D liver segmentation with heterogenous cancer stages in magnetic resonance imaging

PONE-D-21-18482R2

Dear Dr. Onofrey,

We’re pleased to inform you that your manuscript has been judged scientifically suitable for publication and will be formally accepted for publication once it meets all outstanding technical requirements.

Kind regards,

Yan Chai Hum

Academic Editor

PLOS ONE

Additional Editor Comments (optional):

All concerns have been addressed.
---

## [Editor Report · Acceptance letter]

19 Nov 2021

PONE-D-21-18482R2 

Improved performance and consistency of deep learning 3D liver segmentation with heterogenous cancer stages in magnetic resonance imaging 

Dear Dr. Onofrey:

I'm pleased to inform you that your manuscript has been deemed suitable for publication in PLOS ONE. Congratulations! Your manuscript is now with our production department. 

Kind regards, 

on behalf of

Dr. Yan Chai Hum 

Academic Editor

PLOS ONE